# The host cells suppress the proliferation of pseudorabies virus by regulating the PI3K/Akt/mTOR pathway

Lei Xu,[1] Qian Tao,[1] Yang Zhang,[1] Feng-qin Lee,[1] Tong Xu,[1] Li-shuang Deng,[1] Zhi-jie Jian,[1] Jun Zhao,[1] Si-yuan Lai,[1] Yuan-cheng Zhou,[2,3] Ling Zhu,[1] Zhi-wen Xu[1]

**ABSTRACT** Pseudorabies virus (PRV), a member of the alpha-herpesviruses, can infect both the nervous and reproductive systems of pigs, causing neonatal mortality and reproductive failure in sows, which incurs substantial economic losses. Neurotropism is a common characteristic of various viruses, allowing them to cross the blood-brain barrier and access the central nervous system. However, the precise mechanisms by which PRV affects the blood-brain barrier are not well understood. To investigate the mechanism of PRV's interaction with the blood-brain barrier and its engagement with the PI3K/Akt signaling pathway during infection, an *in vitro* monolayer cell model of the blood-brain barrier was established. Our research found that PRV activates Matrix metallopeptidase 2 (MMP2), which degrades Zonula occludens-1 (ZO-1) and consequently enhances the permeability of the blood-brain barrier. PRV infection elevated the transcriptional levels of tissue inhibitor of metalloproteinases 1 (TIMP1) and inhibited its degradation through the ubiquitin-proteasome pathway, leading to higher intracellular concentrations of TIMP1 protein. TIMP1 regulates apoptosis and inhibits PRV replication in mouse brain microvascular endothelial cells through the PI3K/Akt/mTOR signaling pathway. In summary, our study delineates the mechanism through which PRV compromises the blood-brain barrier and provides insights into the host's antiviral defense mechanisms post-infection.

**IMPORTANCE** PRV, known for its neurotropic properties, is capable of inducing severe neuronal damage. Our study discovered that following PRV infection, the expression of MMP2 was upregulated, leading to the degradation of ZO-1. Furthermore, upon PRV infection in the host, the promoter of TIMP1 is significantly activated, resulting in a significant increase in TIMP1 protein levels. This upregulation of TIMP1 inhibits the proliferation of PRV through the PI3K/Akt signaling pathway. This study elucidated the mechanism through which PRV, including the PRV XJ delgE/gI/TK strains, compromises the blood-brain barrier and identifies the antiviral response characterized by the activation of the PI3K/Akt signaling pathway within infected host cells. These findings provide potential therapeutic targets for the clinical management and treatment of PRV.

**KEYWORDS** pseudorabies virus, blood-brain barrier, tight junction proteins, PI3K/Akt, antiviral

Newborn piglets are highly susceptible to PRV, exhibiting severe acute clinical symptoms over a brief period, with a mortality rate that can reach 100% (1). In contrast, fattening pigs that contract PRV often survive but can develop persistent latent infections. In sows, PRV infection in early pregnancy may cause abortion, and in late pregnancy, it can result in stillbirths or mummified fetuses (2, 3). PRV has the ability to cross species barriers and infect a wide range of mammals (4–7). In non-native hosts, the fatality rate of PRV infections can reach up to 100% (8). Recently, a human

**Peer Reviewer** Christopher Aaron Rice, Purdue University, West Lafayette, Indiana, USA

Address correspondence to Zhi-wen Xu, abtcxzw@126.com, or Ling Zhu, abtczl72@126.com.

Lei Xu and Qian Tao contributed equally to this article. The order of authors was decided after consultation between Lei Xu and Qian Tao.

The authors declare no conflict of interest.

See the funding table on p. 12.

endophthalmitis case in China, associated with pig excrement sewage, was traced to a mutant strain of PRV. The subsequent isolation of a PRV variant strain from the patient's cerebrospinal fluid by Liu et al. confirmed the virus's potential to infect humans, causing encephalitis and posing a significant threat to public health safety (9).

Studies have demonstrated that viruses such as West Nile virus and Zika virus can penetrate endothelial cells to access the central nervous system or exploit these cells for systemic spread (10, 11). The central nervous system (CNS) invasion by flaviviruses is often accompanied by immune cell infiltration, indicating a possible "Trojan horse" mechanism for viral entry into the CNS (12). Early infection models in animals revealed a reduction in tight junction proteins within the brains of mice infected with Japanese encephalitis virus, accompanied by increased levels of inflammatory cytokines including IFN-γ, CCL2, TNF-α, and IL-6, which resulted in neuroinflammation and heightened blood-brain barrier (BBB) (13). The strong inflammatory response triggered by viral invasion of the CNS plays a pivotal role in the disruption of the blood-brain barrier. Chen et al. reported that PRV infection in mice can enhance the permeability of the blood-brain barrier (14). However, our preliminary findings indicate that PRV infection might not lead to blood-brain barrier damage in mice (15). However, this outcome requires further confirmation through *in vitro* models. This study aimed to establish an *in vitro* blood-brain barrier model to assess the impact of PRV on the blood-brain barrier and clarify the mechanisms of PRV penetration.

The PI3K/AKT pathway is commonly associated with various human diseases, including cancer, diabetes, cardiovascular diseases, and neurological disorders. It is critical for regulating cellular processes such as apoptosis and autophagy (16). Andrew et al. showed that the PI3K/AKT signaling pathway is activated by US3 protein after PRV infection (17). Our earlier research indicated the activation of the PI3K/AKT signaling pathway in the brains of PRV-infected mice, highlighting its significance in PRV infection (18). Nevertheless, the precise mechanisms underlying its action are yet to be fully understood. This study employed *in vitro* cellular assays to explore the role of PI3K/Akt in PRV infection, including the PRV delgE/gI/TK mutant, aiming to enhance our comprehension of the PRV-host interaction.

## RESULTS

### PRV compromises the integrity of blood-brain barrier by degrading ZO-1

To explore the impact of PRV on the BBB, we utilized an *in vitro* monolayer cell model infected with both the PRV XJ strain and the PRV XJ del gE/gI/TK strain. Following 12 hours of infection, the supernatant was replaced with a fresh culture medium containing 20 µg/mL of fluorescein sodium, and the concentration of fluorescein sodium in the lower chamber of the Transwell was measured. To our surprise, the fluorescein sodium concentration in the lower chamber fluid of both the PRV XJ and PRV XJ del gE/gI/TK groups was markedly higher than that of the mock group, indicating an increase in BBB permeability as a result of PRV infection (Fig. 1A). Furthermore, we detected high viral titers of PRV XJ and PRV XJ del gE/gI/TK in the lower chamber fluid of the Transwells (Fig. 1B). Collectively, these findings strongly suggest that PRV infection disrupts blood-brain barrier permeability.

Tight junction proteins are essential for preserving the blood-brain barrier's integrity by controlling the movement of molecules between endothelial cells. To assess the effects of PRV infection on the blood-brain barrier, we performed experiments measuring ZO-1 expression in bEnd.3 cells. Our results demonstrated significant downregulation of ZO-1 in the PRV XJ and PRV XJ del gE/gI/TK groups compared with the mock group (Fig. 1C and D). Additionally, we noted considerable disruption of intercellular ZO-1 integrity after PRV XJ or PRV XJ del gE/gI/TK infection, in contrast to the mock group, which displayed intact tight junctions (Fig. 1E). Collectively, these results indicate that tight junction proteins degrade following PRV infection.

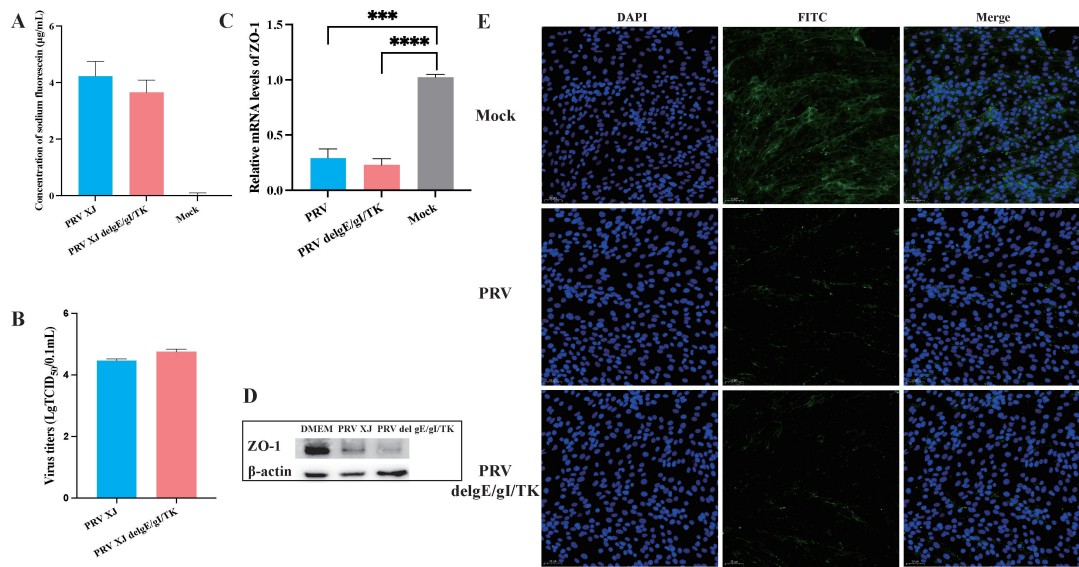

**FIG 1** The effect of pseudorabies virus on the blood-brain barrier. (A) Detection of sodium fluorescein concentration in different infection groups. (B) Determination of viral titers in different infection groups. (C) mRNA expression levels of tight junction proteins ZO-1, occludin-1, and claudin-1. (D) Protein expression level of tight junction protein ZO-1. (E) Integrity of tight junction protein ZO-1 in bEnd.3 cells among different infection groups and control groups. *** represent $P < 0.001$; **** represent $P < 0.0001$.

## PRV degrades ZO-1 through the upregulation of MMP2 expression

Prior research has underscored the importance of MMP2, MMP9, and TIMP1 in preserving the integrity of tight junction proteins and the blood-brain barrier. To further explore the mechanisms behind PRV-induced blood-brain barrier permeability, we assessed the expression levels of MMP2, TIMP1, MMP9, and ZO-1 at various time points post-infection in bEnd.3 cells. Fig. 2A illustrates that MMP2 and TIMP1 mRNA levels were markedly upregulated within 48 h post-PRV infection, whereas ZO-1 mRNA levels progressively declined. Notably, MMP9 levels remained unchanged. Subsequently, western blotting analysis showed an upregulation of MMP2 and TIMP1 proteins post-PRV infection, whereas ZO-1 protein levels were downregulated, and MMP9 expression remained consistent (Fig. 2B). Studies have confirmed the capacity of MMP2 and MMP9 to enhance blood-brain barrier permeability through the degradation of tight junction proteins (19, 20). Additionally, assessments of MMP2 and MMP9 enzyme activities indicated a significant increase in MMP2 activity post-PRV infection, with no notable variation in MMP9 activity (Fig. 2C). To delve deeper into the role of MMP2 in elevating blood-brain barrier permeability, we employed the MMP2 activity inhibitor SB-3CT to suppress its activity. In comparison to the PRV and PRV delgE/gI/TK groups, a significant upregulation of ZO-1 was observed in the SB-3CT-treated counterparts (Fig. 2D). Additionally, blood-brain barrier integrity was improved (Fig. 2E). In the fluorescein sodium permeability assay, we observed that the presence of fluorescein sodium in the SB-3CT treated PRV group and the PRV delgE/gI/TK group was significantly lower compared with that in the respective untreated PRV and PRV delgE/gI/TK groups (Fig. 2F). The findings demonstrate that SB-3CT potently suppresses the PRV-induced and PRV delgE/gI/TK-induced elevation of blood-brain barrier permeability, thereby reinforcing the established regulatory function of MMP2 within this mechanism.

## The host cell prevents the proteasomal degradation of TIMP1

As a matrix metalloproteinase inhibitor, TIMP1 is markedly upregulated in brain microvascular endothelial cells following PRV infection (Fig. 2A). To elucidate the mechanism behind TIMP1 upregulation in PRV infection, the dual-luciferase reporter assay indicated that infections with both PRV and PRV delgE/gI/TK significantly enhanced TIMP1

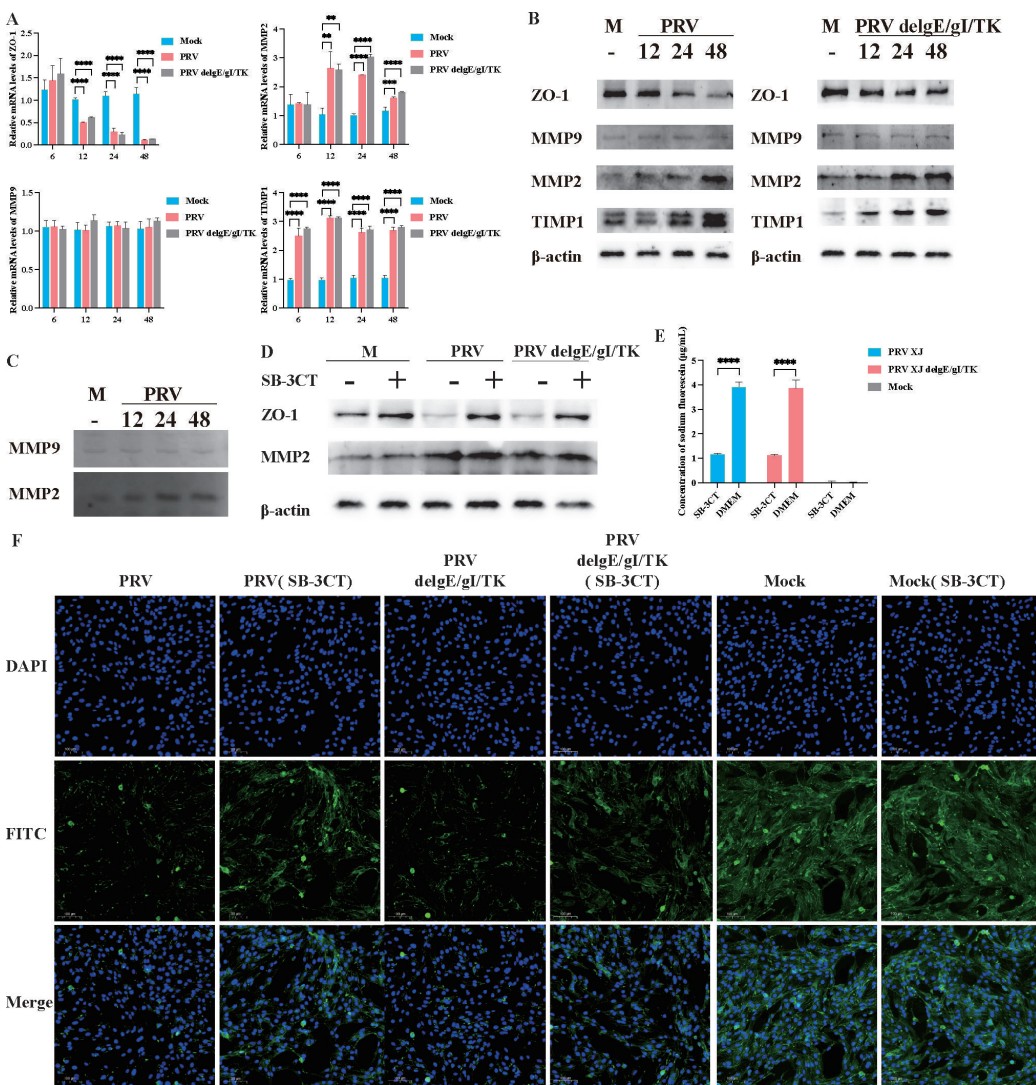

**FIG 2** Mechanisms of pseudorabies virus regulation of tight junction proteins. (A) mRNA expression levels of MMP2, MMP9, TIMP1, and ZO-1 at different time points. (B) Protein expression levels of MMP2, MMP9, TIMP1, and ZO-1 at different time points. (C) Activity assays for MMP9 and MMP2. (D) Protein expression level of ZO-1 after inhibition of MMP2 activity. (E) Detection of sodium fluorescein concentration in different infection groups after inhibition of MMP2 activity. (F) Integrity of tight junction protein ZO-1 in bEnd.3 cells among different infection groups and control groups after inhibition of MMP2 activity. ** represent $P < 0.01$; *** represent $P < 0.001$; **** represent $P < 0.0001$.

promoter activity (Fig. 3A). Subsequently, we treated Bend.3 cells with proteasome inhibitors MG-132 (target the 26Sproteasome) at a concentration of 30 µM and the deubiquitinase inhibitor PR-619 (target USP4, USP8, USP7, USP2, and USP5) at 6.25 µM. Treatment with MG-132 resulted in a significant upregulation of Bcl-2, and no significant differences were observed in the levels of TIMP1, PI3K, p-Akt, and p-mTOR (Fig. 3B). In contrast, PR-619 treatment effectively suppressed the PRV-induced upregulation of TIMP1, PI3K, p-Akt, and p-mTOR (Fig. 3C). Plaque assay and titer detection results confirmed that MG-132-mediated inhibition of TIMP1 degradation significantly reduced the replication of PRV XJ and PRV XJ delgE/gI/TK (Fig. 3D and E). In summary, our findings indicate that PRV infection stimulates the TIMP1 promoter and regulates TIMP1 protein levels via the ubiquitin-proteasome pathway.

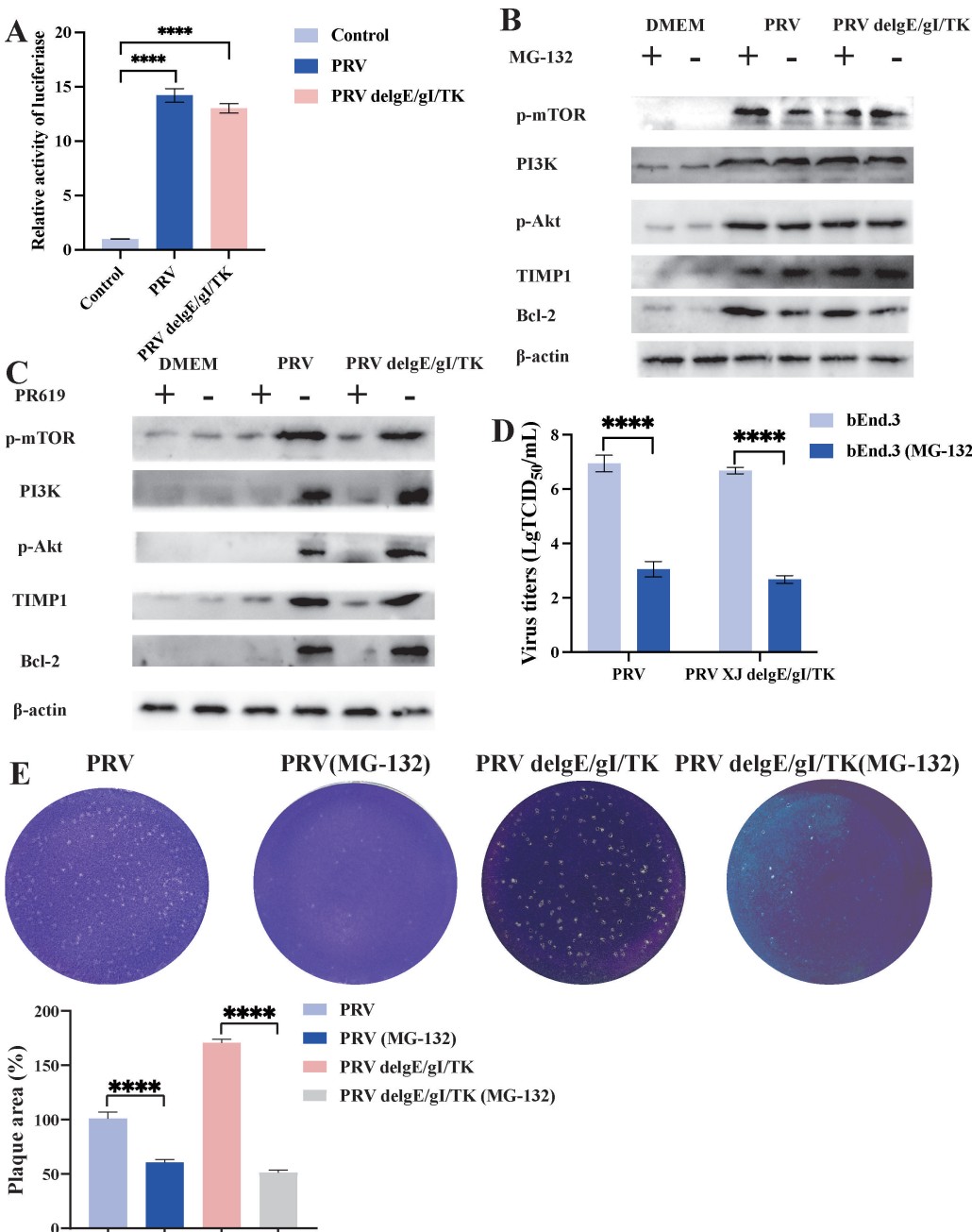

FIG 3 Regulation of TIMP1 mechanism post-host infection with pseudorabies virus. (A) Detection of TIMP1 promoter activity using a luciferase reporter gene assay. (B) The impact of proteasome inhibitors on the TIMP1/PI3K/Akt signaling pathway. (C) The impact of deubiquitinase inhibitors on the TIMP1/PI3K/Akt signaling pathway. (D) Determination of viral titers in different infection groups. (E) PRV plaque assays in different infection groups. **** represent $P < 0.0001$.

## The TIMP1/PI3K/Akt/mTOR signaling pathway is triggered in response to PRV infection

Beyond regulating matrix metalloproteinases, TIMP1 is known to modulate cellular apoptosis through the PI3K/Akt/mTOR signaling pathway (21). To explore the function of TIMP1 in PRV infection, we utilized western blot analysis to assess the activation of the TIMP1/PI3K/Akt/mTOR pathway. Our findings indicate a significant increase in the expression of TIMP1, PI3K, p-Akt, and p-mTOR following PRV infection (Fig. 4A). To investigate whether PRV regulates apoptosis through the PI3K/Akt signaling pathway,

we assessed the expression of p-Akt, Bcl-2, and caspase-3 in bEnd.3 cells treated with 40 µM chlorpromazine (CPZ), beginning 6 h after PRV infection, for a total of 24 h. In PRV-infected bEnd.3 cells treated with CPZ, we observed inhibition of both p-Akt and Bcl-2, in contrast to the upregulation of cleaved-caspase-3. These findings suggest that PRV may regulate apoptosis through the p-Akt/Bcl-2 signaling pathway (Fig. 4B). Furthermore, we found a significant reduction in the activity of cells infected with PRV XJ or PRV XJ delgE/gI/TK and treated with CPZ (Fig. 4C). Fig. 4D and E illustrates that treatment with CPZ results in a significant increase in viral copies for both PRV and PRV XJ delgE/gI/TK strains compared with their respective untreated controls. This finding suggests that CPZ treatment may enhance viral replication, which is an important consideration for understanding the interaction between the virus and the host cell response. Interestingly, the inhibition of Akt by CPZ significantly promoted the proliferation of PRV XJ and PRV XJ delgE/gI/TK (Fig. 4D and E). Collectively, these results indicate that the TIMP/PI3K/Akt signaling pathway is activated post-PRV infection to regulate apoptosis.

## The TIMP1 protein regulates cellular apoptosis and inhibits viral proliferation through the TIMP1/PI3K/Akt/mTOR signaling pathway

To investigate the role of TIMP1 in modulating the PI3K/Akt signaling pathway, we generated a TIMP1-overexpressing cell line and a TIMP1 knockout cell line. Upon TIMP1 overexpression, there was a significant upregulation of PI3K, p-Akt, p-mTOR, and Bcl-2 expression (Fig. 5A). In bEnd.3 cells infected with PRV and PRV delgE/gI/TK, the PI3K/Akt/mTOR signaling pathway was activated, whereas the expression of PI3K, p-Akt, p-mTOR, and Bcl-2 was significantly inhibited in the TIMP1 knockout bEnd.3 cells (Fig. 5B). Overexpression of TIMP1 led to significant inhibition of PRV XJ and PRV XJ delgE/gI/TK proliferation, whereas knockout of TIMP1 resulted in elevated titers for both strains (Fig. 5C). The plaque assay further confirmed that TIMP1 overexpression could inhibit PRV proliferation, and TIMP1 knockout could enhance it (Fig. 5D). These results indicate that PRV or PRV delgE/gI/TK infection activates the TIMP1/PI3K/Akt/mTOR signaling pathway. Altogether, PRV infection upregulates TIMP1 expression, thereby activating the PI3K/Akt/mTOR signaling pathway to modulate cell apoptosis and suppress PRV proliferation.

## DISCUSSION

The BBB is a unique microvascular architecture critical for maintaining cerebral homeostasis. Disruption and increased permeability of the BBB are hallmark features of central nervous system diseases (22). Endothelial cells are tightly connected through tight junction and adherens junction proteins, forming the foundational structure of the BBB. Liu et al. discovered that MMP-2 can degrade occludin, undermining the integrity of the BBB (20). It is well-known that MMP9 plays a pivotal role in regulating BBB integrity through its proteolytic activity, capable of degrading extracellular matrix (ECM) components, facilitating neutrophil migration, and triggering inflammatory responses (19). Wang et al. found that mice lacking MMP9 were protected against traumatic and ischemic brain damage. Similarly, Asahi et al. observed that knocking out MMP-9 effectively alleviated BBB leakage and the decline of ZO-1 in a transient model of middle cerebral artery occlusion, without affecting occluding (23). In our study, MMP9 was expressed at low levels in bEnd.3 cells and was not upregulated following PRV infection. However, MMP2 was significantly upregulated after PRV infection, and treatment with an MMP2 inhibitor (SB-3CT) significantly reduced BBB permeability. These findings align with the results reported by Liu et al (24). Our experimental results suggest that PRV infection compromises the BBB by upregulating MMP2 expression, lowering tight junction protein levels, rather than being mediated by MMP9. Neuroinflammation can trigger elevated levels of MMP2 and MMP9, leading to decreased levels of ZO-1, claudin-5, and occludin (25). Chen Xiangxiu et al. observed a significant upregulation of MMP2 and MMP9 in the brains of PRV-infected mice, along with a notable degradation of ZO-1 (26). The discrepancy in MMP9 expression from our own findings may stem from

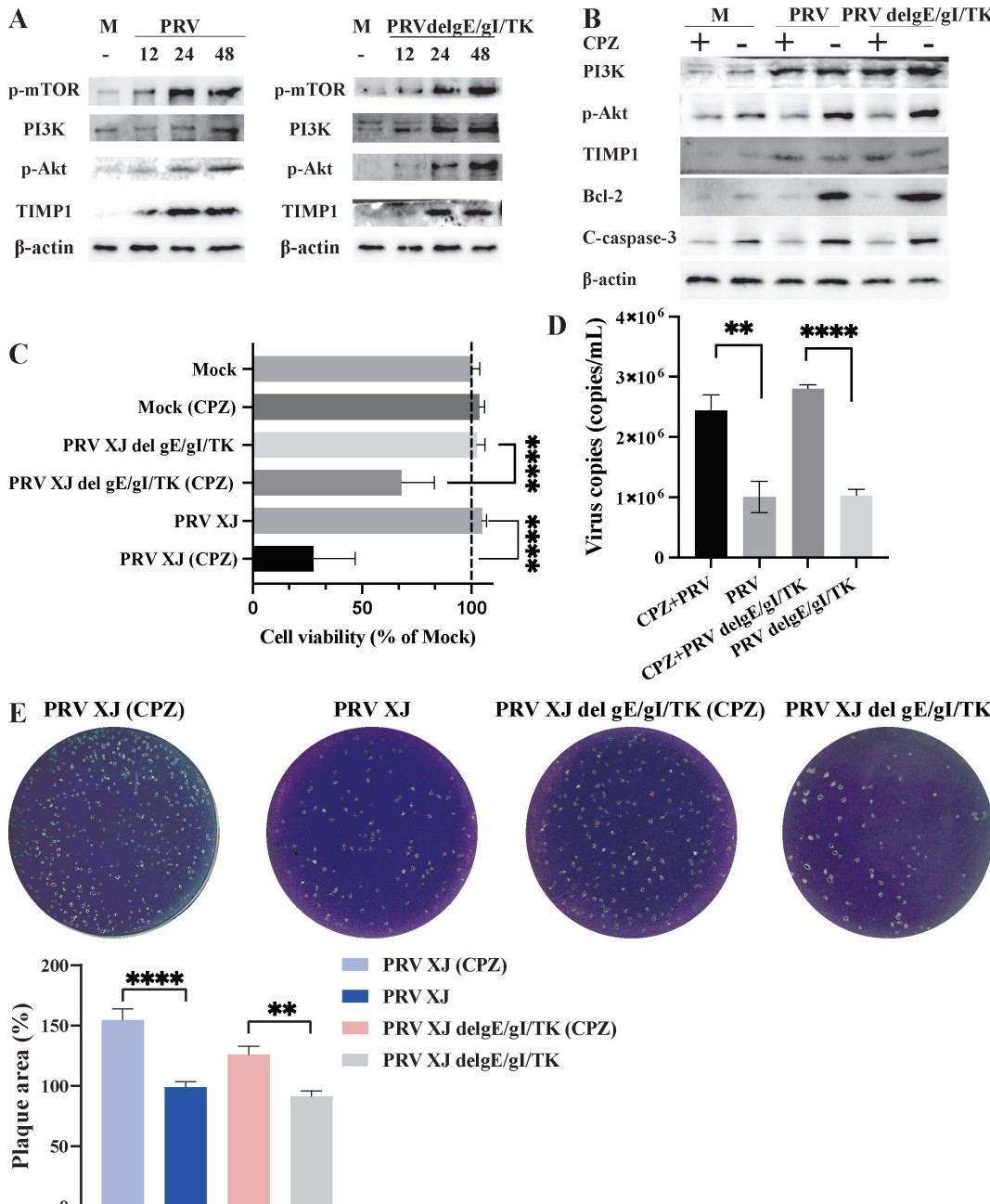

**FIG 4** The function of TIMP1 through the PI3K/Akt signaling pathway. (A) Activation of the PI3K/Akt/mTOR signaling pathway at different time points after PRV infection. (B) Expression of TIMP1, PI3K, Akt, Bcl-2, and cleaved-caspase-3 after inhibition of Akt phosphorylation. (C) Cell viability in different infection groups after inhibition of Akt phosphorylation. (D) Determination of viral titers in different infection groups after inhibition of Akt phosphorylation. (E) PRV plaque assays in different infection groups after inhibition of Akt phosphorylation. ** represent $P < 0.01$; **** represent $P < 0.0001$.

variations in MMP9 expression across tissues and cells. Previous studies have shown that MMP2 is crucial for the maintenance of blood-brain barrier integrity (27). Therefore, we could not further confirm that PRV enhances blood-brain barrier permeability by upregulating MMP2 by knocking out the MMP2 gene.

The TIMP family plays a crucial role in extracellular matrix (ECM) remodeling by acting as endogenous inhibitors that can suppress MMP activity and inhibit integrin-metallo-proteinases. Specifically, TIMP1 can inhibit a wide range of MMPs, including forming a complex with the MMP-9 precursor to inhibit MMP9 activity, and its expression can be induced by various cytokines. Activation of TIMP1 can lead to eosinophilic airway

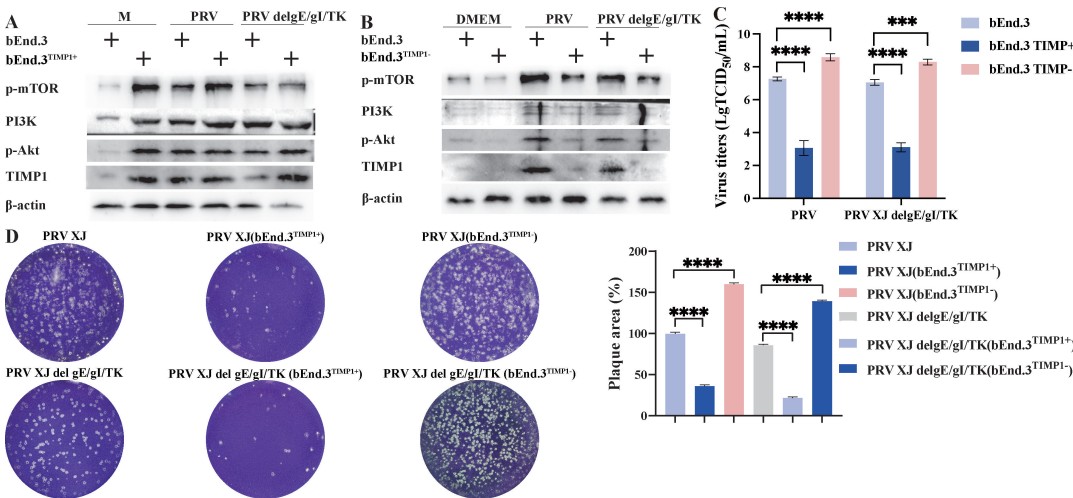

**FIG 5** The Impact of regulating TIMP1/PI3K/Akt on cell apoptosis and virus replication. (A) The effect of overexpressing TIMP1 on the PI3K/Akt/mTOR signaling pathway. (B) The impact of knocking out TIMP1 on the PI3K/Akt/mTOR signaling pathway. (C) Determination of viral titers in different infection groups. (D) PRV plaque assays in different infection groups. **** represent $P < 0.0001$.

inflammation in mice, triggering the activation of eosinophils and macrophages, as well as promoting the release of macrophages toward M2 macrophage polarization, ultimately resulting in type II inflammatory responses (28). In endothelial cells, TIMP-1 can also hinder cell migration by inhibiting the activity of focal adhesion kinase (FAK) (29). Furthermore, TIMP1 is implicated in the regulation of the PI3K/Akt/mTOR pathway, where it inhibits cancer cell apoptosis while promoting cancer cell migration and invasion (21). The phosphoinositide 3-kinases (PI3K)/protein kinase B signaling pathway plays a critical role in inflammatory responses, cell survival, and apoptosis (30–34). PI3K is a crucial downstream effector of both receptor tyrosine kinases and G protein-coupled receptors. It plays a key role in transmitting signals from various growth factors and cytokines through the cell membrane, ultimately activating downstream effectors like AKT and mTORC1. Phosphorylated AKT has been shown to inhibit GSK3 activity, positively regulating downstream target proteins such as c-Myc, Nrf2, and HIF1α, thereby promoting cell growth and preventing apoptosis (35, 36). Previous research has demonstrated that PRV infection can activate the PI3K/Akt signaling pathway to suppress apoptosis (37). Studies have also highlighted the dependence of PRV-induced apoptosis on caspase-3, with inhibition of this process leading to decreased PRV proliferation (38–40). Furthermore, cephalosporin has been found to induce apoptosis in infected cells by targeting the PI3K-Akt and MAPK signaling pathways, consequently reducing HSV-1 infection and replication (41). Bcl-2 is an anti-apoptotic factor and a crucial downstream target of PI3K/AKT-mediated cell survival. The expression of glutathione peroxidase 1 significantly activates Akt phosphorylation, leading to increased Bcl-2 expression and promoting anti-apoptotic effects (42). Our study further validates that following PRV infection, the host triggers the TIMP1/PI3K/AKT/mTOR signaling pathway to increase the expression of Bcl-2, thereby inhibiting cell apoptosis and limiting virus proliferation.

Recent studies have demonstrated that PRV is able to evade the host's innate immune response by targeting critical regulators of protein degradation pathways. UL13 hinders the host's defense mechanisms by degrading antiviral regulator peroxiredoxin 1 (PRDX1) and interferon regulatory factor 3 (IRF3) in a kinase-dependent manner, thereby dampening the innate immune response and reducing the production of type-I interferon (IFN-I) (43, 44). Additionally, UL13 disrupts the stimulator of interferon genes (STING)-mediated signaling pathways by promoting the ubiquitination and degradation of STING through interaction with RING-finger protein 5 (RNF5), ultimately compromising the host's antiviral defenses (45). Conversely, UL24 targets IRF7 and p65 for degradation, counteracting cGAS-STING-mediated IFN-I production and TNF-α-mediated NF-κB

activation, further weakening the host's immune responses against PRV (46). This study has only provided initial confirmation that following PRV infection, the host modulates the level of TIMP1 via the ubiquitin-proteasome pathway, without delving into its underlying mechanism. Further investigation into this aspect will be the focus of our future research.

## MATERIALS AND METHODS

### Cells and viruses

The PRV XJ strain (GenBank accession no. MW893682) was isolated from the brain of a dead piglet that had been vaccinated from a PRV-infected pig farm in Xinjin district, Sichuan province, China, and preserved at our laboratory (the Animal Biotechnology Center at the College of Veterinary Medicine, Sichuan Agricultural University) (47). PRV XJ delgE/gI/TK, which lacks the gE, gI, and TK genes, was previously constructed and stored in our laboratory (15). The bEnd.3 cells were purchased from Yu Chi (Shanghai) Biotechnology Co., Ltd and preserved at our laboratory. The TIMP1 overexpression of bEnd.3 cells and TIMP1 knockout of bEnd.3 cells were constructed and stored in our laboratory. The pGL3 basic vector and pRL-TK vector were purchased from Promega Corporation and preserved at our laboratory. The pGL3-TIMP1-promoter and control pRL-TK vector were stored in our laboratory.

### Quantitative fluorescent PCR and plaque assays

The PRV XJ or the PRV XJ delgE/gI/TK strains were inoculated into bEnd.3 cells in 12-well plates. After incubation at 37°C for 1 h, the viral inoculum was removed. The cells were then washed three times with PBS, followed by the replacement with fresh Dulbecco's Modified Eagle's Medium (DMEM). The cells were further incubated in a cell culture incubator. Samples were collected at 6, 12, 24, and 48 h post-inoculation for RNA extraction and subjected to fluorescent quantitative analysis targeting TIMP1, MMP2, MMP9, and ZO-1.

Genomic DNA was extracted using the Universal Genomic DNA Kit (Cofitt, Jiangsu, China). Quantitative real-time PCR (qRT-PCR) was performed to quantify the DNA copies of the following gene for PRV: gE (forward primer: 5′-CTTCCACTCGCAGCTCTTC T-3′; reverse primer: 5′-TAGATGCAGGGCTCGTACAC-3′). The number of gene copies per microgram of DNA was expressed as $\log_{10}$ copies/µg DNA.

The PRV XJ or PRV XJ delgE/gI/TK were inoculated into BHK-21 cells in 6-well plates, respectively. After incubation at 37°C for 1 h, the viral inoculum was discarded. After washing the cells three times with PBS, they were covered with DMEM containing 1% low melting point agarose. The infected BHK-21 cells were cultured at 37°C with 5% $CO_2$ for 48 h. The supernatant was removed, and the cells were then fixed and stained with formalin-crystal violet staining solution. Finally, pictures were taken for observation. The area of plaques was determined automatically using IPP6.0 software (48).

### The construction of an *in vitro* BBB model and the sodium fluorescein permeability assay

The BBB monolayer Transwell model was utilized in this study. Rat tail collagen was applied to enclose the upper Transwell chamber for 2 h at room temperature. After washing with PBS, DMEM was added to the upper chamber and pre-equilibrated at 37 °C for 2 h. bEnd.3 cells were cultured in the upper chamber with a total volume of 200 µL of DMEM, and 600 µL of DMEM was added to the lower chamber at 37 °C and 5% $CO_2$ for approximately 24 h. The permeability of the BBB monolayer cell model was monitored using sodium fluorescein.

A BBB monolayer cell model was established. The culture medium was discarded and washed three times with PBS. It was inoculated with $10^5$ $TCID_{50}$ of either PRV XJ or PRV XJ dekgE/gI/TK and incubated at 37°C with 5% $CO_2$ for 1 h. The culture medium was

discarded and washed with PBS. Two hundred microliters of fresh DMEM was added to the upper chamber, and 600 µL DMEM, to the lower chamber. Subsequently, DMEM containing 20 µg/mL of sodium fluorescein was added to the upper chamber, and DMEM without sodium fluorescein was added to the lower chamber. It was incubated at 37 °C with 5% CO2 for 4 h, and then, 100 µL of fluid was collected from the lower chamber to measure the $OD_{530}$. The concentration of sodium fluorescein in the fluid from the lower chamber was calculated using the standard curve for sodium fluorescein.

## Indirect immunofluorescence

bEnd.3 cells were seeded onto coverslips, and experimental treatment was conducted after the cells adhered to the surface. The cells were washed three times with PBS preheated to 37°C, followed by fixation with 4% paraformaldehyde at room temperature for 30 min. Wash three times with PBS buffer. Subsequently, permeabilize the membrane with 0.1% TritonX-100 in PBS at room temperature for 5 min. Wash three times with PBS buffer again. The cell slips were blocked in 5% BSA sealing fluid for 1h at room temperature. Subsequently, the primary antibody, ZO-1 polyclonal antibody (21773–1-AP, Proteintech, China), was applied at a dilution of 500 in PBS buffer and incubated overnight at 4°C. After three washes with PBS, the diluted FITC-labeled secondary antibody (511201, Zen-Biosciences, China) was added at a dilution of 2,000, and the slips were incubated at 37°C for 1h. Following another three washes with PBS, the cells were permeabilized with 0.1% Triton X-100 for 10 min, then washed three times with PBS. The slips were mounted using an anti-fade mounting medium containing 4',6-diamidino-2-phenylindole (DAPI) (P0131, Beyotime Biotechnology, China). A fluorescence microscope, NIKON Eclipse ci, was used to observe the cell slips with the following settings: excitation/emission wavelengths for DAPI (Ex/Em 340–380/435–485 nm) and FITC (Ex/Em 465–495/515–560 nm).

## Western blotting

Cells were lysed for protein analysis using RIPA buffer supplemented with protease inhibitors. The protein concentration was quantified using a BCA Protein Assay Kit. Equal amounts of total protein were resolved by sodium dodecyl sulfate-polyacrylamide gel electrophoresis and transferred onto a polyvinylidene fluoride membrane. The membrane blots were saturated with 5% BSA in PBST for 2 h at room temperature and then incubated overnight at 4°C with primary antibodies against PI3K, p-Akt, TIMP1, p-mTOR, Bcl-2, ZO-1, MMP2, MMP9, and β-actin. After incubation, the membrane was washed three times with PBST and incubated with HRP Goat Anti-Rabbit IgG(H + L) or HRP Goat Anti-Mouse IgG(H + L). The signals were visualized with SuperSignalTM West Pico Plus Chemiluminescent Substrate. The gray intensity of proteins was measured using Image J software.

## Effects of metalloproteinase on the permeability of an *in vitro* blood-brain barrier model

Cells were infected with $10^5$ $TCID_{50}$ PRV XJ or $10^5$ $TCID_{50}$ PRV XJ del gE/gI/TK and incubated at 37 °C for 1 h. After incubation, the cells were washed three times with PBS. Subsequently, 2 mL of DMEM containing 2% FBS and 30 µM SB-3CT was added. Following 24 h of cultivation, protein samples were prepared for SDS-PAGE and western blotting analyses to detect the expression levels of MMP2, MMP9, and ZO-1.

In the established in BBB monolayer cell model, cells were inoculated with $10^5$ $TCID_{50}$ PRV XJ or $10^5$ $TCID_{50}$ PRV XJ del gE/gI/TK, incubated for 1 h, and then washed with PBS. 200 µL of DMEM containing 2% FBS and 30 µM SB-3CT was added to the upper chamber, and cultivation was continued for another 24 h. Subsequently, the fluid from the lower chamber was collected and preserved for further use. A sodium fluorescein permeability assay was performed.

## Assay for matrix metalloproteinase activity

Cells were lysed by Radio Immunoprecipitation Assay (RIPA) solution to prepare the protein samples. The protein samples were separated by electrophoresis on polyacrylamide gels containing gelatin (1 mg/mL). The gels were washed twice in 2.5% Triton X-100 to remove SDS and were incubated for 20 h at 37℃ in substrate buffer (50 mmol/L Tris-HCl (pH 8.0), 5 mmol/L CaCl2, and 0.02% NaN3). Gels were stained with Coomassie Brilliant Blue R-250 (Bio-Rad) for 1.5 h and then destained using a mixture of acetic acid, methanol, and water. Observations and photographs were taken using an automatic gel imaging system (49).

## Proteasome and deubiquitinase pathway inhibition experiment

Cells were infected with $10^5$ TCID$_{50}$ PRV XJ or $10^5$ TCID$_{50}$ PRV XJ del gE/gI/TK and incubated at 37 ℃ for 1 h. Then, the cells were treated with 6.25 µM PR-619 or 30 µM MG-132. Following a 24-h incubation period, protein samples were prepared for analysis. SDS-PAGE and western blotting were utilized to assess protein expression levels of TIMP1, PI3K, p-AKT, Bcl-2, and caspase-3.

## Dual-luciferase reporter assay

The bEnd.3 cells were seeded into a 6-well plate and were transfected with pGL3-TIMP1-promoter and control pRL-TK vector for 24 h. The cells were infected with $10^5$ TCID$_{50}$ PRV XJ or $10^5$ TCID$_{50}$ PRV XJ del gE/gI/TK and incubated at 37℃ for 1 h. After a 24-h incubation, luciferase activity was measured using the Dual-Lumi™ luciferase Assay System (RG088S, Beyotime, China) according to the manufacturer's instructions. The relative luciferase activity was obtained by normalizing the firefly luciferase activity against the internal Renilla luciferase control activity.

## Statistical analysis

We performed statistical analysis using GraphPad 7.04 software and conducted one-way analysis of variance (ANOVA). All data are expressed as mean ± standard deviation. A significance level of $P < 0.05$ was considered statistically significant.

### ACKNOWLEDGMENTS

This work was supported by the Chongqing Municipal Technology Innovation and Application Development Project (grant number cstc2021jscx-dxwt BX0007), the Key K&D Program of Sichuan Science and Technology Plan (grant number 2022YFN0007), the Porcine Major Science and Technology Project of Sichuan Science and Technology Plan (grant number 2021ZDZX0010-3), the Sichuan Science and Technology Program Projects (Key R&D Projects) (grant number 2023YFN0021), and the Agricultural Industry Technology System of Sichuan Provincial Department of Agriculture (grant number CARS-SVDIP).

L.X., Z.W.-X., and L.Z. contributed to conceptualization. L.X., Y.Z., Q.T., and T.X. contributed to the formal analysis. L.X., L.S.-D., Y.C.-Z., L.Z., and F.Q.-L. contributed to data curation and original draft preparation. L.X., Z.W.-X., Y.Z., Q.T., T.X., Z.J.-J., J.Z., and Y.T.-Y. contributed to the review and editing. Y.Z., Q.T., L.X., Y.C.-Z., and L.S.-D. contributed to software application. Y.T.-Y., J.Z., Z.J-J., and Y.Z. contributed to supervision. L.Z. contributed to resources. Z.W-X. contributed to validation and resources. Z.W.-X., L.Z., and S.Y-L. contributed to project administration and funding acquisition. All authors have read and agreed to the published version of the manuscript.

### AUTHOR AFFILIATIONS

[1]Key Laboratory of Animal Diseases and Human Health of Sichuan Province, College of Veterinary Medicine, Sichuan Agricultural University, Chengdu, China

²Livestock and Poultry Biological Products Key Laboratory of Sichuan Province, Sichuan Animal Science Academy, Chengdu, China

³Animal Breeding and Genetics Key Laboratory of Sichuan Province, Sichuan Animal Science Academy, Chengdu, China

## AUTHOR ORCIDs

Lei Xu  http://orcid.org/0000-0001-9554-6927
Ling Zhu  http://orcid.org/0000-0002-2682-3465
Zhi-wen Xu  http://orcid.org/0000-0002-9508-6593

## FUNDING

| Funder | Grant(s) | Author(s) |
| --- | --- | --- |
| Chongqing municipal technology innovation and application development project | cstc2021jscx-dxwt BX0007 | Zhiwen Xu |
| Key R&D program of sichuan science and technology Plan | 2022YFN0007, 2023YFN0021 | Zhiwen Xu |
| porcine major science and technology project of sichuan science and technology plan | 2021ZDZX0010-3 | Ling Zhu |
| agricultural industry technology system of sichuan provincial department of agriculture | CARS-SVDIP | Zhi-wen Xu |

## AUTHOR CONTRIBUTIONS

Lei Xu, Conceptualization, Data curation, Formal analysis, Software, Writing – original draft, Writing – review and editing | Qian Tao, Formal analysis, Software, Writing – review and editing | Yang Zhang, Formal analysis, Software, Supervision, Writing – review and editing | Feng-qin Lee, Data curation, Writing – original draft | Tong Xu, Formal analysis, Writing – review and editing | Li-shuang Deng, Data curation, Software, Writing – original draft | Zhi-jie Jian, Supervision, Writing – review and editing | Jun Zhao, Supervision, Writing – review and editing | Si-yuan Lai, Project administration | Yuan-cheng Zhou, Data curation, Software, Writing – original draft | Ling Zhu, Conceptualization, Data curation, Funding acquisition, Project administration, Resources, Writing – original draft | Zhi-wen Xu, Conceptualization, Funding acquisition, Project administration, Resources, Validation, Writing – review and editing

## DATA AVAILABILITY STATEMENT

Generated Statement: The original contributions presented in the study are included in the article/supplementary material, further inquiries can be directed to the corresponding authors.

## ADDITIONAL FILES

The following material is available online.

### Open Peer Review

**PEER REVIEW HISTORY (review-history.pdf).** An accounting of the reviewer comments and feedback.

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
