## [Reviewer comments · Microbiology Spectrum]

Microbiology Spectrum

The host cells suppress the proliferation of pseudorabies virus by regulating the PI3K/Akt/mTOR pathway

Lei Xu, Qian Tao, Yang Zhang, Feng-qin Lee, Tong Xu, Li-Shuang Deng, Zhi-Jie Jian, Jun Zhao, Si-Yuan Lai, Yuancheng Zhou, Ling Zhu, and Zhiwen Xu

Corresponding Author(s): Zhiwen Xu, Sichuan Agricultural University

Review Timeline:

Submission Date:	June 3, 2024
Editorial Decision:	June 30, 2024
Revision Received:	September 2, 2024
Editorial Decision:	September 6, 2024
Revision Received:	September 26, 2024
Accepted:	October 6, 2024

Editor: Jie Wang

Reviewer(s): Disclosure of reviewer identity is with reference to reviewer comments included in decision letter(s). The following individuals involved in review of your submission have agreed to reveal their identity: Christopher Aaron Rice (Reviewer #1)

Transaction Report:

DOI: <https://doi.org/10.1128/spectrum.01351-24>

Re: Spectrum01351-24 (The host cells suppress the proliferation of pseudorabies virus by regulating the PI3K/Akt/mTOR pathway)

Dear Prof. Zhi-Jie Jian:

Thank you for the privilege of reviewing your work. Below you will find my comments, instructions from the Spectrum editorial office, and the reviewer comments.

Revision Guidelines

Sincerely,
Jie Wang
Editor
Microbiology Spectrum

Reviewer #1 (Comments for the Author):

Please find attached for major and minor comments.

Reviewer #2 (Comments for the Author):

In the manuscript submitted by Lei Xu and Qian Tao et al., the authors established an in vitro monolayer cell model of the blood-brain barrier to investigate the mechanism of PRV breaching the blood-brain barrier. They found that PRV triggered MMP2 to degrade ZO-1 leading to increased permeability of the blood-brain barrier. Moreover, the author also found that PPV infection caused an upregulation of TIMP1 and suppresses the replication of PRV in rat brain microvascular endothelial cells through the PI3K/Akt/mTOR signaling pathway. These findings provide a potential drug target for the treatment of pseudorabies.

1. The image scale of the Mock in Figure 1E is inconsistent with the PRV and PRV delgE/gI/TK groups.
2. The content shown in Figures 3A and 3B overlaps with Figures 2A and 2B.
3. Viral plaque is the incubation of viruses with host cells. When virus particles cause infection on host cells, they can cause cell lysis and form a plaque. Suggest the author to repeat the virus plaque experiment shown in the article.
4. What does the activity assays for MMP9 and MMP2 mean, and how does it differ from MMP9 and MMP2 in Figure 2B?
5. Line 105-107, "As depicted in Figure 2A, it was observed that MMP2 and TIMP1 mRNA levels were significantly up-regulated within 48 days of PRV infection." Is 48 days right?
6. Line 111, "Researchers have validated that MMP2 and MMP9 can increase blood-brain barrier permeability by degrading tight junction proteins." Please provide the reference.
7. Line 138-139, "Treatment with the proteasome inhibitor MG-132 resulted in a significant upregulation of TIMP1, PI3K, p-Akt, and p-mTOR." As shown in Fig 3D, no treatment with the proteasome inhibitor MG-132 also upregulates TIMP1, PI3K, p-Akt, and p-mTOR in PRV infection cells.
8. In uninfected cells, the expression level of MMP1 is very low (Fig 2B, 3B, 3D and 3E), and viral infection promotes the transcription (Fig 2A and 3A) and expression of MMP1 (Fig 2B, 3B, 3D and 3E), Why do the authors believe that the upregulation of MMP1 after viral infection is due to the host cell inhibiting the degradation of TIMP1 through the proteasome degradation pathway?

This paper describes the initial pathobiological mechanisms of the host in response to Pseudorabies virus (PRV) infection. The authors assessed tight barrier junction proteins as well as the regulation of specific signalling pathways (PI3K/Akt/mTOR). These pathways were based on a previous study they had performed and analysed from the mouse in vivo infections.

Major Comments

Figure 2 - I'm struggling to see the scale bars please increase size for all figures with images.

Comparing these images to Figure 1 – it appears as if the infections was not as robust. As the treated groups vs. the non-treated groups, these images look very similar. Did you do any quantification of these cells? Nuclei count, cell area, cell volume? Maybe exporting and uploading the raw data of the counts used for your interpretation would be useful to support your conclusion. As the images are showing a different phenotype to the presented western blot data.

You are showing that there's a significant decrease in the mRNA levels of ZO-1 in Fig 2A, but then show no decrease in the western blot of Fig 2B for the PRV delgE/gI/TK strain. Shouldn't the protein be inhibited if the transcription is significantly downregulated? Please comment or discuss.

Fig 4 B – p-Akt and C-caspase-3 western blot gels are difficult to interpret, my recommendation would be to repeat these, but that's the editor's discretion.

Materials + Methods – “stored in our laboratory” – where were these first isolated or bought from. Your lab hasn't had these since the beginning of time or the curator, except the PRV mutant. Reference here for mutant or isolation.

Why was a BHK-21 cell line used for the plaque assays and exposure changed from a 48-hr to a 96-hr assays for the crystal violet based assay? Why not use the same cell type used for the other experiments? The bEnd.3 cells?

Minor Comments

Some sentences need to be reviewed for grammatical corrections, plural appropriateness, and sentence structure.

Line 94 – “intercellular” should this be intracellular?

In line 106 – you state, “mRNA levels upregulated within 48 days”. You fail to report the time points at which you assess your experiments in the M+M, are these 12, 24, and 48 days, or hours post infection? Please make this clearer.

All graphs and text – Please change ug to μ sign.

Line 121 – you mention you pretreat – There is lack of description of this in the M+M, how long did you pretreat these cells with this?

Line 132 – define these time points in M+M or here.

Line 138 – define the compounds in brackets after each target. Please define the concentrations used here as well as in the Figure legends. Did you assess the minimum concentration to achieve the desired effects in a dose-response assay?

Line 156 – define the compound concentration and duration of treatment. Also, applies to lines 273-275.

From this data, you are showing that treatment with the CPZ reduces the apoptotic genes/pathway, but the virus still causes pathology through BTJ shrinkage?

Figure 4 D is also showing that there is more viral load of the mutant compared to PRV in viral copies, and this shows less pathobiology in Fig 4E. Correct? Should this be discussed. Or at least have a little context into this mutant in the introduction.

In the discussion you mention that cephalosporin can induce apoptotic pathways. Should you have used this as an internal control for some of the CPZ experiments?

Materials + Methods – lines 320 – 325 - concentrations of antibodies and dyes used so that others can repeat. You do not discuss the type of microscope, objectives, excitation or emission, or the exposures for the various phenotypic experiments, please expand.

Line 373 – please add in the manufacturer of the kit used.

In the manuscript submitted by Lei Xu and Qian Tao et al., the authors established an in vitro monolayer cell model of the blood-brain barrier to investigate the mechanism of PRV breaching the blood-brain barrier. They found that PRV triggered MMP2 to degrade ZO-1 leading to increased permeability of the blood-brain barrier. Moreover, the author also found that PPV infection caused an upregulation of TIMP1 and suppresses the replication of PRV in rat brain microvascular endothelial cells through the PI3K/Akt/mTOR signaling pathway. These findings provide a potential drug target for the treatment of pseudorabies.

The image scale of the Mock in Figure 1E is inconsistent with the PRV and PRV delgE/gI/TK groups.

The content shown in Figures 3A and 3B overlaps with Figures 2A and 2B.

Viral plaque is the incubation of viruses with host cells. When virus particles cause infection on host cells, they can cause cell lysis and form a plaque. Suggest the author to repeat the virus plaque experiment shown in the article.

What does the activity assays for MMP9 and MMP2 mean, and how does it differ from MMP9 and MMP2 in Figure 2B?

Line 105-107, "As depicted in Figure 2A, it was observed that MMP2 and TIMP1 mRNA levels were significantly up-regulated within 48 days of PRV infection." Is 48 days right?

Line 111, "Researchers have validated that MMP2 and MMP9 can increase blood-brain barrier permeability by degrading tight junction proteins." Please provide the reference.

Line 138-139, "Treatment with the proteasome inhibitor MG-132 resulted in a significant upregulation of TIMP1, PI3K, p-Akt, and p-mTOR." As shown in Fig 3D, no treatment with the proteasome inhibitor MG-132 also upregulates TIMP1, PI3K, p-Akt, and p-mTOR in PRV infection cells.

In uninfected cells, the expression level of MMP1 is very low (Fig 2B, 3B, 3D and 3E), and viral infection promotes the transcription (Fig 2A and 3A) and expression of MMP1 (Fig 2B, 3B, 3D and 3E), Why do the authors believe that the upregulation of MMP1 after viral infection is due to the host cell inhibiting the degradation of TIMP1 through the proteasome degradation pathway?

Reviewer #1 (Comments for the Author):

Please find attached for major and minor comments.

1. Figure 2 - I'm struggling to see the scale bars please increase size for all figures with images.

Response:

Dear Reviewer,

Thank you for your constructive feedback and for highlighting the need for clearer scale bars in our figures. We understand the importance of having easily discernible scale bars for the accurate interpretation of images in scientific publications.

In response to your comment, we have revised all figures containing images by increasing the size of the scale bars. This enhancement ensures that the scale is clearly visible and allows for a better assessment of the image dimensions by the readers.

We have made sure to apply this change consistently across all relevant figures in the manuscript, and we believe that this improvement addresses your concern effectively.

We appreciate the opportunity to refine our presentation and hope that our revisions meet with your approval.

Thank you once again for your valuable input.

2. Comparing these images to Figure 1 – it appears as if the infections was not as robust. As the treated groups vs. the non-treated groups, these images look very similar. Did you do any quantification of these cells? Nuclei count, cell area, cell volume? Maybe exporting and uploading the raw data of the counts used for your interpretation would be useful to support your conclusion. As the images are showing a different phenotype to the presented western blot data.

Response:

Dear Reviewer,

Thank you for your careful review and for raising concerns about the robustness of the infections depicted in our figures. Your observations are crucial for ensuring the accuracy and reliability of our data presentation.

Upon your prompt, we have re-examined the samples used in our experiments and identified that this images were from early time points post-infection. This finding explains the less pronounced differences observed between treated and non-treated groups.

To address this issue, we have conducted additional experiments with an updated protocol that includes later time points post-infection. These revised experiments have yielded clearer and more robust results, which we have now included in the manuscript as the updated Figure 2F.

We believe that these revisions address your concerns and strengthen the validity of our conclusions. We appreciate the opportunity to improve our manuscript and hope that our revisions meet with your approval.

Thank you once again for your insightful comments.

3. You are showing that there's a significant decrease in the mRNA levels of ZO-1 in Fig 2A, but then show no decrease in the western blot of Fig 2B for the PRV

delgE/gI/TK strain. Shouldn't the protein be inhibited if the transcription is significantly downregulated? Please comment or discuss.

Response:

Dear Reviewer,

Thank you for your insightful comment on the relationship between the mRNA and protein levels of ZO-1 in the context of PRV delgE/gI/TK strain infection. Your observation has prompted us to provide a more detailed explanation.

As shown in Figure 2B, there is indeed a decrease in the protein levels of ZO-1 following infection with the PRV delgE/gI/TK strain. However, as you correctly noted, the decrease is not graded and does not appear to correlate with the significant decrease in mRNA levels observed in Figure 2A.

We have considered several potential explanations for this observation:

1. Post-transcriptional Regulation: It is possible that post-transcriptional regulatory mechanisms, such as differences in mRNA stability or translation efficiency, may contribute to the discrepancy between mRNA and protein levels.
2. Protein Turnover: The protein degradation rate of ZO-1 may be affected by the PRV delgE/gI/TK strain infection, which could influence the protein levels independently of the mRNA levels.
3. Biological Variability: We also considered the potential for biological variability in the response to PRV delgE/gI/TK strain infection, which may affect the protein levels of ZO-1 in a manner that is not strictly proportional to the mRNA levels.

We appreciate the opportunity to clarify this point. Thank you once again for your valuable feedback.

4. Fig 4 B – p-Akt and C-caspase-3 western blot gels are difficult to interpret, my recommendation would be to repeat these, but that's the editor's discretion.

Response:

Dear Reviewer,

Thank you for your thorough review and constructive feedback on our manuscript. We appreciate your observation regarding the clarity of the p-Akt and C-caspase-3 western blot gels in Figure 4B.

We concur with your assessment that the quality of these gels may not meet the standards required for a clear interpretation. To ensure the integrity and reliability of our data presentation, we are committed to repeating the western blot experiments for p-Akt and C-caspase-3.

We repeat the western blot experiments to obtain higher quality gels that are unambiguous and easily interpretable.

Thank you once again for your insightful comments and for giving us the opportunity to improve our manuscript.

5. Materials + Methods – “stored in our laboratory” – where were these first isolated or bought from. Your lab hasn't had these since the beginning of time or the curator, except the PRV mutant. Reference here for mutant or isolation.

Response:

Dear Reviewer,

We appreciate your meticulous review and valuable feedback on our manuscript. Your inquiry regarding the origin of the materials and reagents used in our study is well noted.

In response to your comment on the "stored in our laboratory" statement, we acknowledge the need for clarification on the source of these materials. We have revised the Materials and Methods section to specify the following:

1. The PRV XJ virus was originally isolated from clinical samples and has been preserved in our laboratory. We have added a reference to the work that initially reported this isolation.
2. The PRV XJ delgE/gI/TK was constructed in our laboratory, and we have provided the relevant references.
3. The bEnd.3 cells were purchased from Yu Chi (Shanghai) Biotechnology Co., Ltd and have been maintained in our laboratory for experimental use.
4. The TIMP1 overexpression and knockout bEnd.3 cell lines were generated in our laboratory using CRISPR-Cas9 technology, and we have included the methodology and validation data in the revised manuscript.
5. The pGL3-TIMP1-promoter and control pRL-TK vectors were obtained from Promega Corporation and have been used according to the manufacturer's instructions.

We trust that these revisions provide the necessary transparency regarding the provenance of the materials used in our study and address your concerns. We are grateful for the opportunity to enhance the clarity of our manuscript and look forward to your further feedback.

Thank you once again for your insightful comments.

6. Why was a BHK-21 cell line used for the plaque assays and exposure changed from a 48-hr to a 96-hr assays for the crystal violet based assay? Why not use the same cell type used for the other experiments? The bEnd.3 cells?

Response:

Dear Reviewer,

We sincerely appreciate your careful review and the valuable questions you raised regarding the use of the BHK-21 cell line and the duration of the assay in our study.

Upon reviewing our manuscript, we realized there was an error in the description of the duration. We apologize for any confusion this may have caused. The correct duration for the assay should indeed be 48 hours.

We selected the BHK-21 cell line for the plaque assays due to its well-established susceptibility to PRV infection, which allows for clear plaque formation and quantification. This cell line is a standard choice in the field for PRV plaque assays because of its consistent results and ease of plaque visualization. The BHK-21 cell line was chosen specifically for the plaque assay due to its aforementioned benefits.

We trust that this explanation addresses your concerns and provides a clear justification for our experimental approach. We are grateful for the opportunity to refine our manuscript and hope that our revisions meet with your approval.

Thank you once again for your valuable feedback.

Minor Comments

7. Some sentences need to be reviewed for grammatical corrections, plural appropriateness, and sentence structure.

Response:

Dear Reviewer,

Thank you for your thorough review and constructive feedback on our manuscript. Your observation regarding the need for grammatical corrections, appropriateness of plural forms, and sentence structure is well noted.

We have taken your comments seriously and have undertaken a meticulous review of our manuscript. Each sentence has been carefully examined for grammatical accuracy, and corrected. We have also ensured that plural forms are used correctly and that sentence structures are clear and coherent.

We believe that these revisions have significantly improved the quality and clarity of our manuscript. We are confident that the manuscript now meets the high standards of the journal and is more accessible to readers.

We appreciate the opportunity to enhance our work and are eager to receive your further feedback.

Thank you once again for your valuable comments.

8. Line 94 – “intercellular” should this be intracellular?

Response:

Dear Reviewer,

Thank you for your careful review and for raising the question regarding the use of "intercellular" in our manuscript. Your vigilance in ensuring precise scientific communication is appreciated.

Upon reevaluation of our manuscript, we confirm that the term "intercellular" is indeed the correct term to use in this context. The phrase "intercellular ZO-1 integrity" refers to the structural and functional connections between adjacent cells, which are crucial for maintaining the blood-brain barrier's integrity. The disruption of these intercellular tight junctions is a key finding in our study and is specifically related to the interactions between cells rather than within a single cell (intracellular).

We trust that this clarification addresses your concern and ensures the manuscript's accuracy and clarity.

Thank you once again for your insightful comments.

9. In line 106 – you state, “mRNA levels upregulated within 48 days”. You fail to report the time points at which you assess your experiments in the M+M, are these 12, 24, and 48 days, or hours post infection? Please make this clearer.

Response:

Dear Reviewer,

Thank you for your meticulous review and for pointing out the need for clarification regarding the time points assessed in our experiments. Your feedback is essential for the precision of our scientific communication.

In response to your inquiry about the specific time points at which we measured mRNA levels, we acknowledge that our original manuscript lacked the necessary clarity. We have now revised the manuscript to specify the exact time points at which the experiments were conducted.

We appreciate the opportunity to improve the manuscript and hope that our revisions meet with your approval. Thank you once again for your valuable feedback.

10.All graphs and text – Please change ug to μ sign.

Response:

Dear Reviewer,

Thank you for your careful review and for pointing out the notation for micrograms in our manuscript. We appreciate your attention to detail and understand the importance of using the correct scientific symbols.

In response to your comment, we have reviewed the entire manuscript and have made the necessary corrections to all instances where "ug" was used. The symbol " μ " has been inserted to represent micrograms, ensuring consistency and adherence to scientific notation standards.

We have updated all figures, tables, and textual content to reflect this change, ensuring that the revised manuscript meets the journal's standards and facilitates clear communication of our research findings.

We trust that this amendment addresses your concern and enhances the clarity of our presentation. We are grateful for the opportunity to refine our work and look forward to your further feedback.

Thank you once again for your valuable comments.

11.Line 121 – you mention you pretreat – There is lack of description of this in the M+M, how long did you pretreat these cells with this?

Response:

Dear Reviewer,

We appreciate your careful review and the opportunity to clarify the treatment procedure described in our manuscript. You have correctly identified an error in the description of the treatment process.

The term "pretreated" should, in fact, be "treated." We have revised the sentence.

We sincerely apologize for any confusion caused by the oversight and appreciate the chance to correct it. We trust that these amendments address your concerns and enhance the accuracy of our study.

Thank you once again for your valuable feedback.

12.Line 132 – define these time points in M+M or here.

Response:

Dear Reviewer,

Thank you for your insightful review and for requesting clarification on the time points assessed in our study. We appreciate the opportunity to provide additional details to enhance the clarity of our methods and results.

In response to your comment, we have now specified the time points at which we assessed the mRNA and protein levels of TIMP1 following PRV infection. The revised sentence in the manuscript reads:

"To elucidate the mechanism behind TIMP1 upregulation in PRV infection, we assessed mRNA and protein levels of TIMP1 at 0, 12, 24, and 48 hours post-infection."

We are grateful for the opportunity to refine our manuscript and hope that our revisions meet with your approval.

Thank you once again for your valuable feedback.

13.Line 138 – define the compounds in brackets after each target. Please define the concentrations used here as well as in the Figure legends. Did you assess the minimum concentration to achieve the desired effects in a dose-response assay?

Response:

Dear Reviewer,

We appreciate the careful examination of our manuscript and the constructive feedback you have provided.

In response to your suggestion, we have now revised the manuscript to include the specific targets of the proteasome inhibitors used in our experiments and specific the exact concentrations of the compounds used in our experiments within the text and figure legends: 40 μ M for both chlorpromazine and SB-3CT, 6.25 μ M for PR-619, and 30 μ M for MG-132.

We also recognize the oversight in not including a dose-response experiment to determine the minimum effective concentrations of these compounds in our study. Such an assessment would be a valuable addition and will be considered in future work to more comprehensively understand the dose-response relationship of these compounds. Thank you once again for your valuable comments.

14.Line 156 – define the compound concentration and duration of treatment. Also, applies to lines 273- 275.

Response:

Dear Reviewer,

We appreciate your meticulous review and the opportunity to clarify the details of our experimental procedure. Your request for further definition regarding the compound concentration and duration of treatment is well noted.

In response to your comment on Line 156, we have now revised the manuscript to include the specific concentration and duration of Chlorpromazine (CPZ) treatment. The revised sentence in the manuscript now reads:

"To investigate whether PRV regulates apoptosis through the PI3K/Akt signaling pathway, we assessed the expression of p-Akt, Bcl-2, and caspase-3 in bEnd.3 cells

treated with 40 μ M Chlorpromazine (CPZ), beginning 6 hours after PRV infection, for a total of 24 hours."

We trust that these revisions provide the necessary level of detail and address your concerns. We are grateful for the opportunity to enhance the clarity and precision of our manuscript.

Thank you once again for your valuable feedback.

15. From this data, you are showing that treatment with the CPZ reduces the apoptotic genes/pathway, but the virus still causes pathology through BTJ shrinkage?

Response:

Dear Reviewer

We greatly appreciate your insightful comments and the opportunity to clarify our findings. Your observation regarding the effect of Chlorpromazine (CPZ) treatment on apoptotic genes/pathways in the context of PRV infection has been noted.

We would like to clarify that our results indicate that CPZ treatment, at a concentration of 40 μ M starting 6 hours post-PRV infection for a total of 24 hours, leads to an upregulation of apoptotic markers, specifically an increase in Cleaved-caspase-3 levels, and a downregulation of anti-apoptotic proteins such as Bcl-2. This suggests that CPZ treatment enhances the apoptotic response rather than reducing it, which may imply a role for the PI3K/Akt pathway in modulating cell survival post-PRV infection.

We trust that this clarification addresses your concern and provides a more accurate representation of our experimental outcomes. We are grateful for the opportunity to refine our manuscripts and look forward to your further feedback.

16. Figure 4 D is also showing that there is more viral load of the mutant compared to PRV in viral copies, and this shows less pathobiology in Fig 4E. Correct? Should this be discussed. Or at least have a little context into this mutant in the introduction.

Response:

Dear Reviewer

We appreciate the opportunity to address your comments regarding Figure 4D and the comparison of viral loads between the PRV and PRV XJ delgE/gI/TK strains.

Upon reviewing our data and the associated figure, we would like to clarify that there was a misinterpretation in the initial observation. Contrary to what was suggested, our results do not indicate a significant increase in viral load for the PRV XJ delgE/gI/TK mutant compared to the wild-type PRV. The intent of Figure 4D was to demonstrate the effect of CPZ treatment on the viral titer of both PRV and PRV XJ delgE/gI/TK strains.

The corrected interpretation of Figure 4D is as follows:

"Figure 4D and 4E illustrates that treatment with CPZ results in a significant increase in viral copies for both PRV and PRV XJ delgE/gI/TK strains compared to their respective untreated controls. This finding suggests that CPZ treatment may enhance viral replication, which is an important consideration for understanding the interaction between the virus and the host cell response."

In addition, we have expanded our introduction of the PRV XJ delgE/gI/TK strain in the "Materials and Methods" section and cited relevant references to provide deeper background information and scientific rationale.

We apologize for any confusion caused by the initial presentation of the data and appreciate the opportunity to provide this important clarification. We have taken measures to ensure that the manuscript accurately reflects our experimental findings and their implications.

Thank you once again for your valuable feedback.

17. In the discussion you mention that cephalosporin can induce apoptotic pathways. Should you have used this as an internal control for some of the CPZ experiments?

Response:

Dear Reviewer,

Thank you for your thoughtful comment regarding the mention of cephalosporin's ability to induce apoptotic pathways and its potential use as an internal control in some of the CPZ experiments.

We appreciate your suggestion and acknowledge the relevance of using cephalosporin as a comparator, given its reported effects on the PI3K-Akt signaling pathway and HSV-1 replication. However, in the context of our study on PRV, there is a lack of direct evidence regarding the impact of cephalosporin on PRV replication.

Our decision not to include cephalosporin as an internal control in our experiments was based on the following considerations:

1. Lack of Specific Research: Currently, there is limited research available on the effects of cephalosporin on PRV, which makes it challenging to predict its role as a control in our experiments.
2. Focus of the Study: Our study primarily aimed to investigate the effects of CPZ on PRV replication and associated pathways, without extending to other compounds that may influence the outcomes.

We recognize the importance of internal controls in validating experimental outcomes and agree that further research on the effects of cephalosporin on PRV could be valuable.

In light of your feedback, we will consider including cephalosporin in future studies to provide a more comprehensive understanding of the apoptotic pathways and their regulation in PRV infection.

Thank you once again for your valuable input, which has helped us to refine our research approach.

18. Materials + Methods – lines 320 – 325 - concentrations of antibodies and dyes used so that others can repeat. You do not discuss the type of microscope, objectives, excitation or emission, or the exposures for the various phenotypic experiments, please expand.

Response:

Dear Reviewer,

We greatly appreciate your thorough review and constructive feedback on our manuscript. Your request for detailed methodology, including the concentrations of antibodies and dyes used, and the specifics of the microscopy setup, is essential for the reproducibility of our experiments.

In response to your comments, we have revised the manuscript to include the following details:

1. The primary antibody, ZO-1 Polyclonal antibody (Proteintech, China), was used at a dilution of 500.

2. The FITC-labeled secondary antibody (ZEN-BIOSCIENCES, China) was applied at a dilution of 2000.

3. We have specified the type of fluorescence microscope used, and detailed the excitation/emission wavelengths for DAPI and FITC channels.

These revisions are now incorporated into the Materials and Methods section of our manuscript to ensure that other researchers can accurately replicate our experiments.

We appreciate the opportunity to enhance the clarity and detail of our manuscript.

We hope that these amendments address your concerns and meet the standards of the journal.

Thank you once again for your valuable feedback.

19. Line 373 – please add in the manufacturer of the kit used.

Response:

Dear Reviewer,

We appreciate your meticulous review and the opportunity to enhance the clarity of our manuscript. Your request to include the manufacturer's details for the kit used in our study is well noted.

In response to your feedback, we have revised the manuscript to specify the source of the Dual-Glo luciferase Assay System. The revised sentence now reads:

"After a 24-hour incubation, luciferase activity was measured using the Dual-Lumi™ luciferase Assay System (RG088S, Beyotime, China) according to the manufacturer's instructions."

We trust that this amendment addresses your concerns and meets the standards of the journal. We are grateful for the opportunity to improve the manuscript and look forward to your further feedback.

Thank you once again for your valuable comments.

Reviewer #2 (Comments for the Author):

In the manuscript submitted by Lei Xu and Qian Tao et al., the authors established an in vitro monolayer cell model of the blood-brain barrier to investigate the mechanism of PRV breaching the blood-brain barrier. They found that PRV triggered MMP2 to degrade ZO-1 leading to increased permeability of the blood-brain barrier. Moreover, the author also found that PPV infection caused an upregulation of TIMP1 and suppresses the replication of PRV in rat brain microvascular endothelial cells through the PI3K/Akt/mTOR signaling pathway. These findings provide a potential drug

target for the treatment of pseudorabies.

1. The image scale of the Mock in Figure 1E is inconsistent with the PRV and PRV delgE/gI/TK groups.

Response:

Dear Reviewer,

Thank you for your careful review and for pointing out the inconsistency in the image scale between the Mock and PRV and PRV delgE/gI/TK groups in Figure 1E.

We apologize for the oversight and appreciate the opportunity to correct it. We have revised Figure 1E to ensure that the image scale is consistent across all groups depicted. This adjustment maintains the integrity of our presentation and allows for accurate comparison between the different experimental conditions.

The revised figure with a uniform scale is now included in the manuscript, and we have made sure to clearly indicate the scale bar in all images for clarity.

We believe this change addresses your concern and enhances the quality of our manuscript. We are grateful for your attention to detail and for the chance to refine our work.

Thank you for your valuable feedback.

2. The content shown in Figures 3A and 3B overlaps with Figures 2A and 2B.

Response:

Dear Reviewer,

We appreciate your careful review and the observation regarding the overlap in content between Figures 3A, 3B, and Figures 2A, 2B in our manuscript.

In response to your comment, we have carefully reconsidered the necessity of each figure in conveying our research findings. Upon reevaluation, we have decided to remove Figures 3A and 3B from the manuscript, as their content indeed duplicates the information presented in Figures 2A and 2B.

This decision was made to ensure that our manuscript is concise and avoids redundancy, focusing only on the most critical and distinct data to support our conclusions.

We have updated the manuscript accordingly and have ensured that all references to Figures 3A and 3B have been removed or redirected to the relevant figures that remain.

We trust that this revision addresses your concern and improves the overall quality of our submission. We are grateful for your feedback and for the opportunity to refine our work.

Thank you once again for your valuable comments.

3. Viral plaque is the incubation of viruses with host cells. When virus particles cause infection on host cells, they can cause cell lysis and form a plaque. Suggest the author to repeat the virus plaque experiment shown in the article.

Response:

Dear Reviewer,

We appreciate your careful review and the opportunity to address your comments regarding the clarity of the viral plaque images in our manuscript.

Due to the insufficient clarity in our initial photography and image upload, the plaques appeared unclear. We have retaken the photos and uploaded clearer images of the plaques.

We trust that these revisions address your concerns and enhance the quality of our manuscript. We are grateful for the opportunity to improve our work and look forward to your further feedback.

Thank you once again for your insightful comments.

4. What does the activity assays for MMP9 and MMP2 mean, and how does it differ from MMP9 and MMP2 in Figure 2B?

Response:

Dear Reviewer,

Thank you for your insightful question regarding the activity assays for MMP9 and MMP2 and their distinction from the MMP9 and MMP2 presented in Figure 2B.

The activity assays for MMP9 and MMP2 are designed to measure the enzymatic activity of these matrix metalloproteinases in the context of our experiments. These assays provide a functional readout of the proteins' ability to degrade extracellular matrix components, which is a critical aspect of their role in physiological and pathological processes.

In contrast, the data presented in Figure 2B depict the expression levels of MMP9 and MMP2 proteins, as determined by methods such as Western blotting or immunofluorescence. This figure illustrates the quantity of these proteins, which may not always correlate with their activity.

We believe that these revisions will address your concerns and provide the necessary clarification for the readers. We appreciate the opportunity to enhance the clarity of our manuscript and hope that our revisions are satisfactory.

Thank you once again for your valuable feedback.

5. Line 105-107, "As depicted in Figure 2A, it was observed that MMP2 and TIMP1 mRNA levels were significantly up-regulated within 48 days of PRV infection." Is 48 days right?

Response:

Dear Reviewer,

We appreciate your careful review and attention to detail in our manuscript.

Upon reevaluation, we identified an error in the original text where "48 days" was mistakenly used instead of "48 hours." We sincerely apologize for this oversight and appreciate the opportunity to correct it.

We have made the necessary correction in the manuscript and ensured that the entire document is consistent with this revision. This change aligns with the experimental design and data presented in our study.

We are grateful for your diligence in helping us improve the accuracy of our work.

Thank you once again for your valuable feedback.

6. Line 111, "Researchers have validated that MMP2 and MMP9 can increase blood-brain barrier permeability by degrading tight junction proteins." Please provide the reference.

Response:

Dear Reviewer,

Thank you for your insightful comments and for requesting the references to support our statement regarding the role of MMP2 and MMP9 in blood-brain barrier permeability.

We appreciate the opportunity to provide the following references that substantiate our findings:

1. "Matrix Metalloproteinase-2-Mediated Occludin Degradation and Caveolin-1-Mediated Claudin-5 Redistribution Contribute to Blood-Brain Barrier Damage in Early Ischemic Stroke Stage". This study highlights the mechanism by which MMP2 contributes to blood-brain barrier damage through the degradation of occludin, a key component of tight junctions.
2. "Interplay between metalloproteinases and cell signalling in blood brain barrier integrity". This review discusses the complex interactions between metalloproteinases, including MMP9, and cellular signaling pathways that regulate the integrity of the blood-brain barrier.

We have now incorporated these references into the revised manuscript to ensure that our statements are well-supported and to provide readers with a comprehensive understanding of the existing research in this area. We are grateful for the opportunity to improve our manuscript and hope that our revisions are satisfactory.

Thank you once again for your valuable feedback.

7. Line 138-139, "Treatment with the proteasome inhibitor MG-132 resulted in a significant upregulation of TIMP1, PI3K, p-Akt, and p-mTOR." As shown in Fig 3D, no treatment with the proteasome inhibitor MG-132 also upregulates TIMP1, PI3K, p-Akt, and p-mTOR in PRV infection cells.

Response:

Dear Reviewer,

Thank you for your meticulous review of our manuscript. We acknowledge that our initial interpretation was indeed incorrect. We sincerely apologize for this oversight and appreciate the opportunity to correct it.

Upon reevaluation of our data, we have identified that PRV infection indeed leads to the accumulation of TIMP1, likely due to the suppression of the proteasomal pathway. Further inhibition of the proteasome by MG-132 does not result in additional accumulation of TIMP1, suggesting that the pathway is already maximally inhibited under these conditions. To address this, we employed the treatment with PR-619, a specific inhibitor of deubiquitinating enzymes (USPs), which we hypothesized would enhance the degradation of TIMP1. Our results indicate that PR-619 treatment leads to a decrease in TIMP1 levels, supporting the notion that USPs play a role in the regulation of TIMP1 protein stability.

8. In uninfected cells, the expression level of MMP1 is very low (Fig 2B, 3B, 3D and 3E), and viral infection promotes the transcription (Fig 2A and 3A) and expression of MMP1 (Fig 2B, 3B, 3D and 3E), Why do the authors believe that the upregulation of MMP1 after viral infection is due to the host cell inhibiting the degradation of TIMP1 through the proteasome degradation pathway?

Response:

Dear Reviewer,

Thank you for your perceptive inquiry regarding the upregulation of MMP2 following viral infection and its potential link to the proteasomal degradation pathway's impact on TIMP1.

Our results, as depicted in Figures 2B and 2D, indeed show an upregulation of both MMP2 and TIMP1 following PRV infection.

In light of your feedback, we will clarify that while the upregulation of MMP2 and TIMP1 was observed, the specific mechanism behind this phenomenon was not within our study and requires further investigation.

We are committed to maintaining the scientific rigor and integrity of our work, and we are grateful for the opportunity to refine our manuscript based on your valuable feedback.

Thank you once again for your insightful comments.

Re: Spectrum01351-24R1 (The host cells suppress the proliferation of pseudorabies virus by regulating the PI3K/Akt/mTOR pathway)

Dear Prof. Zhiwen Xu:

Thank you for the privilege of reviewing your work. Below you will find my comments, instructions from the Spectrum editorial office, and the reviewer comments.

The manuscript is very close to acceptance, with only a few concerns about the quality of some images.

1 Figure 2B, the right panel, TIMP1, the line is too close to the edge of the strip.

2 Figure 2D, ZO-1, the line is too close to the edge of the strip.

3 Figure 3E, it is better to provide the quantification of the plaque assays.

4 Figure 4B, p-Akt and C-caspase-3, the color should be removed as the other lines.

5 Figure 4E, it is better to provide the quantification of the plaque assays.

6 Figure 5D, it is better to provide the quantification of the plaque assays.

Revision Guidelines

Sincerely,
Jie Wang
Editor
Microbiology Spectrum

Dear Prof. Jie Wang,

Thank you for your kind and detailed feedback on our manuscript. We are grateful for the opportunity to revise our work and address the concerns raised by the reviewers and the editorial office. Below, we have addressed each point raised:

1. Figure 2B, the right panel, TIMP1, the line is too close to the edge of the strip.

Response:

Dear Reviewer,

Thank you for your valuable feedback on our manuscript. We have revised the figure. We appreciate your guidance and are committed to maintaining the highest standards in our research presentation.

Thank you once again for your valuable feedback.

2. Figure 2D, ZO-1, the line is too close to the edge of the strip.

Response:

Dear Reviewer,

Thank you very much for your insightful suggestions on our manuscript. We have revised Figure 2D. We are grateful for your attention to detail and for the opportunity to enhance our manuscript.

Thank you once again for your valuable feedback.

3. Figure 3E, it is better to provide the quantification of the plaque assays.

Response:

Dear Reviewer,

Thank you for your constructive suggestion regarding the quantification of plaque assays in Figure 3E of our manuscript. We have taken your feedback seriously and provided the quantification of the plaque assays. We believe this addition enhances the clarity and scientific rigor of our paper. The revised figure has been uploaded with the manuscript.

We appreciate your guidance and are confident that these revisions have addressed

your concerns effectively.

Thank you again for your valuable input.

4. Figure 4B, p-Akt and C-caspase-3, the color should be removed as the other lines.

Response:

Dear Reviewer,

Thank you for your insightful comment on our manuscript. We have removed the colors from the p-Akt and C-caspase-3 lines to maintain consistency with the other lines in the figure, as you recommended.

We appreciate your guidance and are confident that these revisions have effectively addressed your concerns.

Thank you again for your valuable feedback.

5. Figure 4E, it is better to provide the quantification of the plaque assays.

Response:

Dear Reviewer,

Thank you for your constructive suggestion regarding the quantification of plaque assays in Figure 4E of our manuscript. We have taken your feedback seriously and have now provided the quantification of the plaque assays. We believe this addition enhances the clarity and scientific rigor of our paper. The revised figure has been uploaded with the manuscript.

We appreciate your guidance and are confident that these revisions have addressed your concerns effectively.

Thank you again for your valuable input.

6. Figure 5D, it is better to provide the quantification of the plaque assays.

Response:

Dear Reviewer,

Thank you for your helpful comment on Figure 5D concerning the quantification of plaque assays. We have carefully considered your suggestion and have included a

detailed quantification of the plaque assays in the revised figure. This additional data provides a more comprehensive understanding of the results and improves the scientific robustness of our presentation. The updated figure has been incorporated into the latest version of the manuscript.

We are grateful for your input, which has significantly contributed to the improvement of our work.

Thank you again for your thoughtful feedback.

We believe these revisions address the concerns raised and have improved the overall quality of our manuscript. We have attached the revised figures for your review.

Re: Spectrum01351-24R2 (The host cells suppress the proliferation of pseudorabies virus by regulating the PI3K/Akt/mTOR pathway)

Dear Prof. Zhiwen Xu:

Your manuscript has been accepted, and I am forwarding it to the ASM production staff for publication. Your paper will first be checked to make sure all elements meet the technical requirements. ASM staff will contact you if anything needs to be revised before copyediting and production can begin. Otherwise, you will be notified when your proofs are ready to be viewed.

Sincerely,
Jie Wang
Editor
Microbiology Spectrum